# The Use of Microsensors to Assess the Daily Wear Time of Removable Orthodontic Appliances: A Prospective Cohort Study

**DOI:** 10.3390/s22072435

**Published:** 2022-03-22

**Authors:** Marek Nahajowski, Joanna Lis, Michał Sarul

**Affiliations:** 1Department of Integrated Dentistry, Wroclaw Medical University, Krakowska 26, 50-425 Wroclaw, Poland; michal.sarul@umw.edu.pl; 2Clinic of Orthodontics, Academic Policlinic of Stomatology, Krakowska 26, 50-425 Wroclaw, Poland; joanna.lis@umw.edu.pl; 3Department of Orthodontics and Dentofacial Orthopedics, Wroclaw Medical University, Krakowska 26, 50-425 Wroclaw, Poland; 4Clinic of Integrated Dentistry, Academic Policlinic of Stomatology, Krakowska 26, 50-425 Wroclaw, Poland

**Keywords:** microsensors, compliance, orthodontics

## Abstract

Orthodontic treatment with removable appliances is still common in children and adolescents. However, their effectiveness depends primarily on the patients’ compliance. Currently, it is possible to check the daily wear time (DWT) of the removable appliances using special microsensors. The aim of this prospective cohort study was to assess the degree of patients’ compliance depending on the type of removable appliance used. In total, 167 patients (87 F, 80 M) were enrolled in the study and were treated with block appliances (Klammt, Twin-Block), Schwarz plates, and block appliances in combination with headgear. All patients were followed up for 6 months with the mean daily wear time checked at followup visits using TheraMon^®^ microsensors fitted in the appliances. It has been shown that the type of appliance influences the patients’ compliance. The DWT for the Twin Block was significantly longer compared to the DWT for the other appliances. Girls have been shown to wear removable appliances better than boys. It has been proven that the majority of patients do not follow the orthodontist’s recommendations, wearing removable appliances for just over half of the recommended time. Microsensors can be used for objective verification of patients’ compliance, which allows for a reliable assessment of the effectiveness of treatment with removable appliances.

## 1. Introduction

Removable orthodontic appliances have been widely used since the first half of the 20th century, when Andresen and Schwarz introduced a monoblock appliance and an active plate into orthodontics [1]. Since then, many authors have improved these appliances, adapting them to the treatment of different malocclusions. As fixed orthodontic appliances are now in common use, standard removable appliances may seem like a relic of a bygone era; however, they have their undeniable advantages [2]. Removable orthodontic appliances are easy to manufacture and use, show resistance to damage, and reduce the risk of caries development during orthodontic treatment. Above all, they are inexpensive and are ideal for solving many orthodontic issues in early and interceptive treatment, i.e., in general treatment of children and adolescents [3,4]. The largest disadvantage related to using removable appliances is the difficulty in prediction and monitoring the patient’s compliance during treatment, while it is clear that these appliances must be worn as recommended by the orthodontist to be effective.

In the last century and even at the beginning of the present one, a fully objective assessment of the compliance of orthodontic patients treated with different types of removable appliances was virtually impossible. This has affected not only clinical procedures but also the reliability of various studies related to this type of therapy [5]. Patient adherence is a factor that, if ignored, may have a significant effect on studies concerning the effectiveness of removable orthodontic appliances, thereby affecting the treatment strategy recommendations based on these studies [6].

Now, this problem has been solved by electronic systems that monitor the daily wear time (DWT) of orthodontic appliances. TheraMon^®^ (MC Technology GmbH, Hargelsberg, Austria) is one such system, which is very effective clinically [6]. TheraMon^®^ consists of (a) polyurethane-coated sensors measuring 12.8 × 8.7 × 4.2 mm that read and record temperature every 15 min to an accuracy of 0.1 °C; (b) a docking station that reads the data stored in the sensors; and (c) software that not only enables an analysis, visualization, and interpretation of data but also identifies attempts of tampering/cheating by patients. As proven, DWT values recorded by TheraMon^®^ microsensors were found to be underestimated by merely 4% [7]. Therefore, these sensors, except for being easy to use, have proven to be reliable and accurate in assessing DWT of orthodontic appliances [6].

The microsensors mounted in removable appliances have been shown to be reliable predictors of good patient cooperation [8,9,10,11,12]. However, there is still a lack of information regarding which type of removable orthodontic appliance is best tolerated by patients and, thus, enables good patient compliance.

The study aims to investigate whether the motivation to continue orthodontic treatment with removable appliances, as expressed by the quality of patient’s compliance, is dependent on the type of orthodontic appliance.

## 2. Materials and Methods

The approval of the Bioethics Committee No. KB-322/2014 (Bioethics Committee of Wroclaw Medical University) was obtained prior to the study.

### 2.1. Sample Size Calculation

The study enrolled 167 patients (80 boys and 87 girls). The mean age of participants was 9.4–11.8 years (10.3 years on average). The sample size was calculated to provide 80% power to identify a 20% difference between groups, *p* < 0.05.

### 2.2. Study Design

Inclusion criteria included healthy patients without clefts, systemic diseases, or previous orthodontic treatment.

Depending on their malocclusion, patients were eligible for treatment with a removable appliance, i.e., an orthodontic appliance that is completely dependent on compliance, selecting one of three methods of treatment:(1)Functional treatment using modifications to the monoblock appliance;(2)Active treatment using a lower or upper Schwarz appliance (S); or(3)Functional active treatment using the twin block appliance combined with headgear (TB + HG).

The twin block (TB) or Klammt (K) appliances were randomly used for functional treatment.

The following study groups were obtained according to the type of removable orthodontic appliance used:

TB (*n* = 53), K (*n* = 53), S (*n* = 39), and TB + HG (*n* = 22). 

Highly trained dental technicians fitted the TheraMon^®^ sensors to acrylic plates of the removable appliances in accordance with the manufacturer’s recommendations. The sensors were entirely covered with acrylic, which prevented them from coming into direct contact with the oral environment (Figure 1).

As stated in the EC-Declaration of Conformity of Medical Device, Medical Device Directive 2007/47/EG, the TheraMon^®^ sensors did not complicate the design of the orthodontic appliance and did not affect the comfort of use. The sensor recorded the temperature every 15 min for up to 18 months. The recorded time when the temperature detected by the sensor exceeded 35 °C corresponded to the time when an activator appliance was worn. A special integrated circuit with a 16-kilobyte internal, electrically erasable, programmable read-only memory was used for recording the data. These data were read using a TheraMon^®^ station to generate diagrams with DWT information (Figure 2). 

Each TheraMon^®^ sensor was activated when the orthodontic appliance was given to the patient. Patients and their parents signed an informed consent for participation in the study after receiving information about: (a) the future presence of a microsensor in the appliance, (b) complete harmlessness of the sensors, and (c) the rules for voluntary participation in the study and the possibility to cease participation anytime.

All participants were advised to wear their orthodontic appliance continuously for a minimum of 12 h per day and to have regular monthly checkups. The therapy was provided by orthodontists who were previously trained in the use of a computer program used for reading the sensor data. The sensor data were read at each followup visit. The TheraMon^®^ software automatically calculated the DWT for each patient.

### 2.3. Statistical Analysis

The obtained data were statistically analyzed using Statistica v.13.3 (TIBCO Software Inc., Palo Alto, CA, USA). The Shapiro-Wilk test was used for assessing the normality of the empirical distributions, while the Brown-Forsythe test was used for determining the homogeneity of the variance within subgroups, assuming a test significance level of *p* < 0.05. The relationship between DWT and patient gender/the type of appliance used was analyzed using the Student’s *t*-tests and the analysis of variance (ANOVA).

## 3. Results

The empirical DWT distributions in individual subgroups did not differ significantly from the normal distribution (*p* > 0.05). The assumptions of the applicability of the Brown-Forsythe analysis of variance were met in the studied groups.

The mean followup time was 6 months. The DWT varied between 0.34 h/day (a female patient treated with TB + HG) and 21.9 h/day (a male patient treated with TB). Seven patients treated with a Klammt appliance and four patients treated with a Schwarz appliance did not attend any followup visit, making it impossible to read the sensors. Only seven patients (six girls and one boy) complied with the medical recommendations (minimum 12 h of continuous wear per day). The average DWT in each group by gender is shown in Table 1. 

There was a statistically significant correlation between patients’ adherence and their gender. The real DWT of orthodontic appliances was longer in girls compared to boys by an average of 1.3 h (7.1 vs. 5.8 h; *p* = 0.014; Figure 3). 

It was also found that the type of orthodontic appliance had an effect on patient compliance. The DWT of the TB was significantly longer compared to the DWT of the other three types of orthodontic appliances (*p* < 0.05; Figure 4). The DWT of K, S, and TB combined with HG were not significantly different (*p* > 0.05).

## 4. Discussion

In recent years, the use of microsensors has enabled clinicians to objectively monitor the DWT of removable orthodontic appliances during treatment. Studies using the TheraMon^®^ system reveal that patient compliance is much weaker than that required by orthodontists [6,8,9,10,11,12,13]. Moreover, in this study, DWT averaged 7.1 h per day in girls and 5.8 h in boys, with a minimum of 12 h of DWT recommended. This is consistent with studies by other authors in which patient compliance never exceeds 7–9 h out of the recommended 8–15 h per day, [6,8,9,10,11,12] indicating that it is only possible to be confident in patients’ wearing their orthodontic appliances overnight. In previous studies concerning patient compliance during treatment with removable orthodontic appliances, it was proved that although the average DWT was 63–67% of the recommended 14–15 h per day, this percentage revealed a wide range in individual patients: 0.0–89.3% [8,10,11]. Schaefer et al. [10] found that patient compliance was close to the required values (i.e., more than 12 h per day) in 7% of patients, while Schott and Ludwig [8] stressed that 25% of patients wore their orthodontic appliance significantly less than 7 h per day, which significantly reduced the chance of clinical success, i.e., successful malocclusion treatment. 

Schott et al. [9] found comparably low patient compliance in those treated with functional appliances and those wearing retainers. Although Sergl and Zentner [14] found that the degree of patient compliance depended on the type of an orthodontic appliance, DWT was not objectively verified in their study. The sensor-based monitoring of patient compliance increases the reliability of the results of this study. There was no statistically significant difference in terms of the patients’ compliance treated with K, S, and TB combined with HG. However, the DWT of TB was significantly longer (Figure 4), which may indicate that patients become easily accustomed to this appliance; thus, it can be concluded that TB is potentially a highly effective orthodontic appliance in terms of treatment effects. It can also be assumed that the construction bite does not hinder the patients’ acceptance of their orthodontic appliance; thus, it does not affect the level of patient compliance, which is also evidenced by the relatively long DWT.

The least compliant patients were those treated with TB combined with HG (TB + HG group). The significant difference in terms of DWT compared to the TB group may indicate patients’ reluctance to wear HG. 

Previous studies concerning the patients’ compliance treated with HG [15,16,17] revealed that the average DWT was 5–7 h per day compared to the recommended DWT of 12 h. These values did not change even though patients were aware that they were being monitored. The results obtained in the current study (5.8 ± 1.1 h/day for girls and 5.2 ± 0.9 h/day for boys, which corresponds to 43–48% of the recommended 12 h per day) are very similar to the values reported in the literature and to the average DWT = 5.8 h, which was identified in the systematic review by Al-Moghrabi et al. [18].

Our study reveals that the average DWT of orthodontic appliances exceeded 7 h/day in the K and TB groups, was approximately 6.5 h in the S group, and did not exceed 5.5 h/day in the TB + HG group. Only seven patients from all groups fully complied with the doctor’s recommendations. These results are similar to those obtained in a similar study by Al-Kurwi et al. [19]. There are several plausible explanations for these results. Previous studies found that decreased quality of life due to malocclusion and dental appearance is related to the cooperation of adolescent patients [20,21]. Patients’ motivation, in addition to the influence of their peers and authority figures, was found to be a decisive factor in terms of adherence to treatment recommendations [21,22]. It was found that patients treated in private medical facilities followed recommendations regarding DWT much more strictly than those treated under compulsory health insurance [10]. 

There are contradictory opinions concerning the effect of gender on patient compliance [10,11,16,23,24]. The current study reported statistically significant better compliance from girls. 

Although patients knew that their compliance was monitored with a microsensor, most of them did not achieve the recommended DWT of 12 h. These findings are consistent with the results of previous studies that revealed that patients’ compliance is insufficient, even when patients and parents are aware that the DWT is recorded [11]. This is consistent with results obtained by other authors [12,25]. Importantly, by collecting sensor data it was proven, as in studies by other authors [5,14], that patients usually do not change their behavior during treatment. Therefore, since patient motivation during therapy is not successful, it is relevant to initially select only those patients who will comply well.

It is advised that the DWT of removable orthodontic appliances be 12–14 h. Unfortunately, our study proved that the objectively verified DWT in question was less than 7 h on average. Moreover, even in the most compliant group—girls treated with TB, the DWT was on average 8.1 h/day (Table 1). The fact that almost 7% of patients in this study were completely uncompliant also compromises the prognosis for successful treatment with a removable orthodontic appliance.

This leaves no doubt that previous assumptions regarding treatment efficacy are overestimated, as measuring and monitoring DWT was subject to a high risk of bias [26]. These assumptions contradict the highly reliable result of our study, which clearly justifies the need to re-evaluate the effectiveness of removable orthodontic appliances in order to update the outdated recommendations concerning their DWT. However, such an evaluation requires only cooperative patients to be eligible for the study. Their simple selection is facilitated by the conclusions drawn, among others, in our previous studies concerning the influence of treatment needs and individual patients’ perception of smile attractiveness on their compliance during treatment [27,28].

The analysis of the degree of patients’ compliance during treatment is a complex problem, at the borderline between psychology and medicine; the very definition of compliance is also controversial. There is not complete agreement on the meaning of the term itself. However, regardless of the definition, patient compliance is crucial to the success of orthodontic treatment, especially with removable appliances. Many factors can influence the degree of patient compliance. Therefore, many researchers have focused on establishing these factors, which would make it possible to predict patient compliance before an orthodontic appliance is designed and manufactured [8,29,30]. Our previous studies [27] as well as the study by Amado et al. [31] found that the degree of patient compliance depends on personality traits of patients and, more importantly, their parents.

On the other hand, the study by Daniels et al. emphasized the motivation of the patient and their parents to enter treatment, as this is a factor which significantly influences the subsequent compliance [22]. In terms of at least two-step therapy, Bos reveals that the success with which the first phase of orthodontic therapy is completed plays a key role in tendency of patients for better compliance in subsequent stages [16]. A specific example of the phase 1 orthodontic treatment is functional therapy. According to the available scientific data [1,32], treatment of certain malocclusions, e.g., Class II, is most effective during the period of growth spurt, puberty. Unfortunately, Albino et al. [30] confirmed the clinical observations of many orthodontists that it is much more difficult to motivate adolescent patients than adults. Moreover, Dinwiddie and Müller prove that children’s compliance weakens with the onset of puberty [32]. According to Tsomos et al., [11] large-scale studies are needed to establish the correlation between patient age and compliance. In such a perspective, an objective evaluation of methods for assessing patient compliance that is a natural consequence of a patient’s motivation to use removable orthodontic appliances seems quite relevant.

The current lack of objective evaluation means that the degrees of craniofacial bone growth modification reported in studies involving patients treated with functional therapy can be controversial. The question arises as to whether the results would have been different if the patients’ compliance monitoring had been fully dependable. A subjective and medical interview-based assessment of the degree of patients’ compliance during functional therapy is a fundamental limitation not only of simple original research or comparative studies but also of randomized trials. The studies by Ghafari et al. and O’Brien et al. indicate statistically significant differences in terms of cephalometric measurements of patients treated and untreated for class II malocclusion [33,34,35]. Proffit questions the relevance of these results to the clinical effectiveness of functional therapy. He stresses that in view of the impossibility of objective evaluation of the degree of patients’ compliance, all existing studies concerning the effectiveness of functional therapy, which reveal the full potential of therapeutic options of functional appliances, are not fully dependable. Our recently published studies prove that very satisfactory results of functional treatment can be obtained in compliant patients [36].

The degree of the discipline of the patients depends little on the severity of the malocclusion as measured by IOTN (Index of Orthodontic Treatment Need) [27]. By planning early orthodontic treatment with removable appliances, it can be assumed that patients with mild malocclusion will be less compliant, which should influence health care providers’ decisions to limit the public funds spent on treating such disorders. Unfortunately, patients’ compliance is unpredictable in those with severe malocclusion, which means that treatment should be carefully and objectively monitored and discontinued if the orthodontist’s recommendations are not followed.

This article is a summary of a pilot study; however, as it continues in our university, it may provide important evidence as to whether confrontation of patients with uncontested confirmation of their degree of cooperation is likely to influence the therapeutic outcomes achieved. It is also significant that all study participants benefited from orthodontic treatment reimbursed by the National Health Fund. 

The results of this study are consistent with the findings of previous studies [11,37], in the sense that the observed large individual variation in terms of DWT highlights the need for the adolescent patient to be actively involved in treatment. Recording the DWT of removable orthodontic appliances using microsensors is a useful tool in the early detection of non-compliant patients, which enables rapid intervention to improve patients’ compliance. 

### Limitations

Age at treatment onset and malocclusion severity were not randomized within various groups of patients treated with different types of orthodontic appliances, which may have biased the data obtained and, thus, negatively affected the results.

As the observation period covered the summer months, a typical decrease in patients’ compliance could be observed during holidays (mainly in July and August). It should not be surprising that patients are significantly less motivated to adhere to treatment recommendations during leisure and holidays. Some of them completely discontinued treatment, especially patients treated with headgear (HG). Therefore, monitoring of patients over a longer period of time could reduce this problem.

## 5. Conclusions

(1)Children treated under compulsory health insurance wear removable orthodontic appliances for much shorter periods of time than recommended; very poor patient compliance, nearly 54% of the required 12 h per day, probably significantly reduces the effectiveness of orthodontic treatment.(2)Since patients treated with removable appliances are most willing to use a twin block appliance (TB), this appliance is most often chosen for functional treatment in orthodontics.(3)Further research should focus on how best to encourage patients to adhere to treatment recommendations in order to increase the effectiveness of orthodontic treatment with removable appliances.(4)Microsensors are a valuable tool that allows for the verification of previously conducted research and the conclusions resulting therefrom but also for carrying out research that was once impossible, which is of key importance for the development of orthodontics in the future.

## Figures and Tables

**Figure 1 sensors-22-02435-f001:**
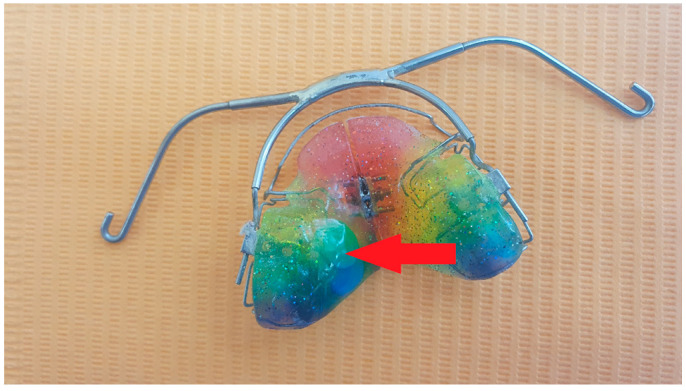
Removable orthodontic appliance with TheraMon^®^ sensor embedded in acrylic (arrow).

**Figure 2 sensors-22-02435-f002:**
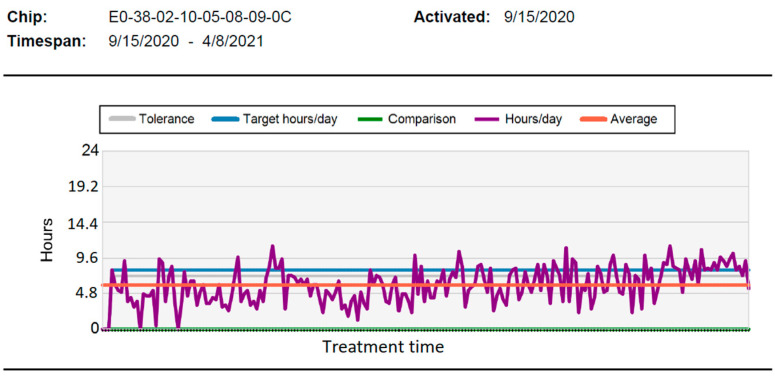
Example graph that illustrates the average DWT of orthodontic appliances, which was recorded by the TheraMon^®^ sensor and was automatically generated by the software Legend: tolerance: deviation from average wearing time; target h/day: recommended wearing time; comparison: an option to compare the results of different patients, not used here; h/day: graph showing the actual wearing time; and average: mean daily wear time.

**Figure 3 sensors-22-02435-f003:**
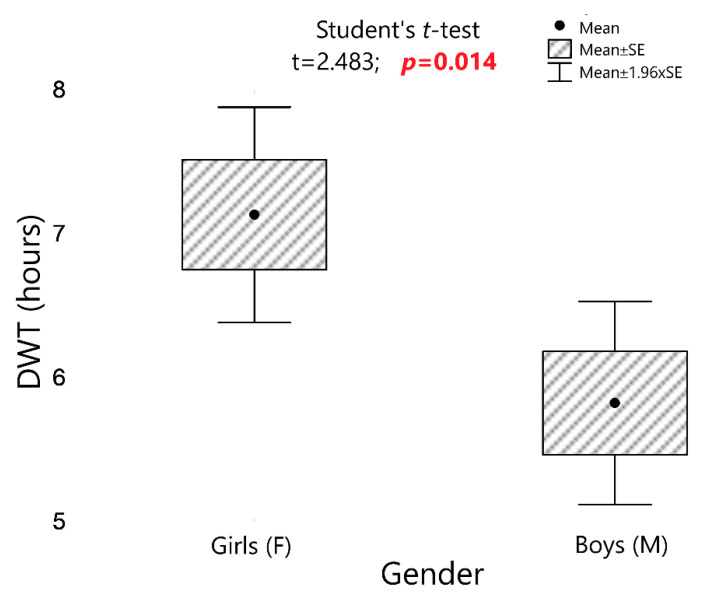
The real DWT of orthodontic appliances for girls and boys and the results of the significance test. DWT: daily wear time.

**Figure 4 sensors-22-02435-f004:**
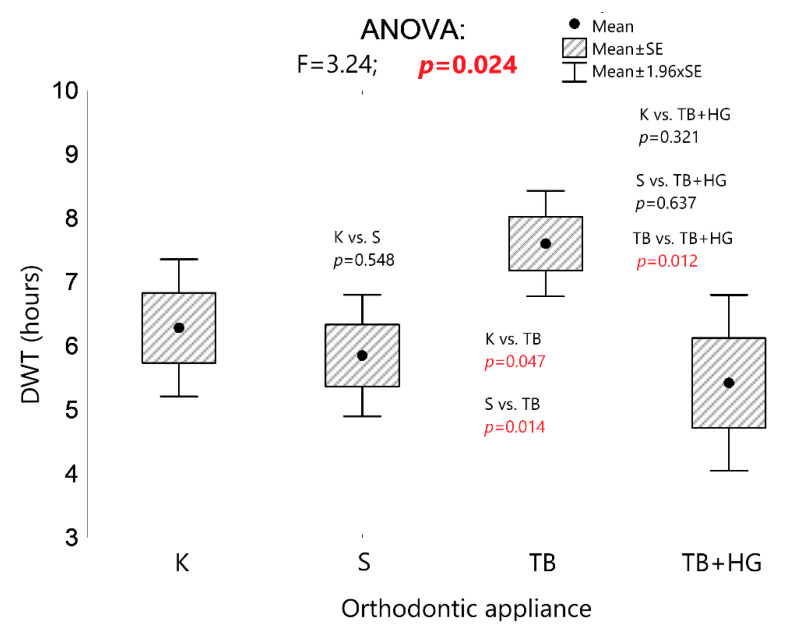
The univariate analysis of variance of the real DWT of orthodontic appliances in groups of patients that differ in terms of the type of orthodontic appliance, and the results of the univariate analysis of variance and post hoc tests (least significant difference test; LSD test). K: Klammt appliance; S: Schwarz appliance; TB: twin block appliance; TB+HG: twin block appliance combined with headgear.

**Table 1 sensors-22-02435-t001:** Descriptive statistics of the real DWT of orthodontic appliances.

Study Group/Type of Appliance	Gender	*N*	Mean ± SE	95% CI
K	F	29	7.2 ± 0.6	6.6–7.8
K	M	22	5.1 ± 0.7	4.4–5.8
S	F	21	6.4 ± 0.7	5.6–7.1
S	M	18	5.3 ± 0.8	4.5–6.1
TB	F	28	8.1 ± 0.6	7.4–8.7
TB	M	27	7.1 ± 0.6	6.5–7.8
TB+HG	F	9	5.8 ± 1.1	4.7–6.9
TB+HG	M	13	5.2 ± 0.9	4.2–6.1

SE—standard error of the mean, 95% CI—95% confidence interval for the mean.

## Data Availability

The datasets used and/or analyzed during the current study are available from the corresponding author on reasonable request.

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
