# Peer review of "The Use of Microsensors to Assess the Daily Wear Time of Removable Orthodontic Appliances: A Prospective Cohort Study"

_sensors, 2022, doi:10.3390/s22072435_

Round 1

Reviewer 1 Report

In this paper, the authors presented a study entitled “The use of microsensors to assess the daily wear time of removable orthodontic appliances” with aim  to investigate the motivation to continue orthodontic treatment with removable appliances, as expressed by the quality of patient’s compliance, is dependent on the type of an orthodontic appliance. In general, the manuscript is very interesting . The topic is in line with the journal aim. An English-language review is required.

My recommendations are the following:

Please insert on the title and abstract the type of the study in order to be immediately understandable for the reader. The introduction section is clear but on the other hand up to line 50 there is little bibliography(Only 2 references)

Material and Methods: This section is not  clear. Please organize this section with subgroups: Study design, Sample size calculation, statistical analysis

Discussion: Is well-performed and in line with results.

The conclusions are not clear and immediate, rewrite this paragraph by simplifying the conclusions into a few paragraphs.

According to this Reviewer’s consideration, novelty and quality of the paper, publication of the present manuscript is recommended after minor revision.

Reviewer 2 Report

This manuscript described a clinical study to investigate patient compliance in orthodontic treatment, using microsensors embedded in removable appliances to check the daily wear time. This real-world study shows the feasibility of using microsensors as an objective measurement of patients’ compliance, which may facilitate precise evaluation and treatment planning for orthodontists. The study is well-executed and the manuscript is clearly written.

I have only a few comments for the authors:

  1. Fig.1: Many readers of the journal have a non-dental background. I wonder if it would be clearer to indicate where the sensor is in the picture, e.g., with an arrow.
  2. Fig.2: This figure needs revisions/caption. It is not clear what the x axis means. Also, the line types in the figure legend (in the black box) do not match the actual lines used in the graph. Some parameters, such as “tolerance”, “comparison”, and “average” require a definition.
  3. The reviewer is not an expert in statistics, but I wonder if the authors are sure about the normality of the data as well as the homogeneity of population variance. These results were not reported in the manuscript. Therefore, it is impossible to determine whether the data meet the assumptions for the selected statistical tests.

Reviewer 3 Report

This is a well conducted and presented study on the use of microsensors embedded in the removable orthodontic appliances with the purpose of objectifying the time of wear in a group of patients of mean age 10.3 yrs and the therapy costs covered by National Health Insurance.

Thera are only few minor issues to be considered.

ABSTRACT

Line 15 introduce abbreviation in brackets for ''daily wear time''

RESULTS

Figures should be described in more details, including legends for the types of appliances K, S, TB, TB+HG and DWT (Figure 3 and 4)  

CONCLUSIONS

This section is too long and some parts are not supported by the results. Third and fourth paragraph (line 329-338) should be removed.
